# Using ADAS to Future-Proof Roads—Comparison of Fog Line Detection from an In-Vehicle Camera and Mobile Retroreflectometer

**DOI:** 10.3390/s21051737

**Published:** 2021-03-03

**Authors:** Ane Dalsnes Storsæter, Kelly Pitera, Edward McCormack

**Affiliations:** 1Norwegian Public Roads Administration (NPRA), Directorate of Public Roads, 7030 Trondheim, Norway; 2Department of Civil and Environmental Engineering, Norwegian University of Science and Technology (NTNU), 7491 Trondheim, Norway; kelly.pitera@ntnu.no (K.P.); edm@uw.edu (E.M.); 3Civil and Environmental Engineering, University of Washington, Seattle, WA 98195, USA

**Keywords:** lane detection, retroreflectometer, road asset management, road maintenance, ADAS, automated driving, road infrastructure

## Abstract

Pavement markings are used to convey positioning information to both humans and automated driving systems. As automated driving is increasingly being adopted to support safety, it is important to understand how successfully sensor systems can interpret these markings. In this effort, an in-vehicle lane departure warning system was compared to data collected simultaneously from an externally mounted mobile retroreflectometer. The test, performed over 200 km of driving on three different routes in variable lighting conditions and road classes found that, depending on conditions, the retroreflectometer could predict whether the car’s lane departure systems would detect markings in 92% to 98% of cases. The test demonstrated that automated driving systems can be used to monitor the state of pavement markings and can provide input on how to design and maintain road infrastructure to support automated driving features. Since data about the condition of lane marking from multiple lane departure warning systems (crowd-sourced data) can provide input into the pavement marking management systems operated by many road owners, these findings also indicate that these automated driving sensors have an important role in enhancing the maintenance of pavement markings.

## 1. Introduction

Road authorities around the world face the challenges of maintaining visible road markings under varying conditions. There are a number of factors that affect the degradation of road markings, including the material (thermoplastic, spray plastic, paint, etc.), location/climate (coastal, inland, etc.), share of studded tires usage, annual average daily traffic (AADT), pavement surface characteristics and conditions, heavy vehicle percentages, quality control in applying the marking material and the use of salts, abrasives or mechanical snow removal [1,2,3]. 

Gathering data to keep an updated status and inventory of road marking quality is a time-consuming process using specialized equipment and personnel [4]. Road asset management systems are being developed to help with these tasks. A Nordic certification system for road marking materials was introduced in 2015 and adopted by Norway, Sweden, and Denmark [5]. The certification system is based on the European standards EN 1824 Road marking materials—Road trials, EN 436 Road marking materials—Road marking performance for road users, and EN 12,802 Road marking materials—Laboratory methods for identification. Documented performance measurements of material samples applied on test fields on public roads are the basis for the certification system. Performance requirements include the coefficient of retroreflected luminance (R_L_), under wet and dry conditions, luminance coefficient under diffuse illumination (Q_d_) as well as friction and color coordinates. Approval is granted in relation to the number of wheel passages the material will withstand [5]. 

In the United States, the Missouri Department of Transportation developed a Pavement Marking Management System (PMMS) as “the only practical method to allow state highway agencies to track the materials, age, cumulative traffic exposure, and retroreflectivity level of existing pavement markings and to enable systematic decisions concerning when, and with what materials, existing markings should be renewed or replaced.” [6]. Likewise, between June 2001 and December 2002 a maintenance management program for the North Dakota Department of Transportation was developed. The program investigated the use of technology such as cameras and positioning devices, along with new software, to allow for more efficient registration of road signs in the field [4]. 

Advanced driver assistance systems (ADAS) have been developed to support human drivers. Lane departure warning (LDW) systems, a form of ADAS, have the potential to decrease the number of accidents but typically rely on road markings to do so [7,8]. The increasing use of cameras and automation in ADAS introduces a possibility to gain access to data for road asset management systems through crowdsourcing [9]. ADAS could thus provide quality assurance systems such as the PMMS with the data to successfully predict maintenance needs.

A report from the European Road Assessment Programme (EuroRAP) has suggested that inadequate road maintenance and the lack of marking consistency across Europe negatively influence the efficacy and implementation of advanced driver assistance systems [10]. ADAS functionality is a step toward increasing driving automation, introducing a new road user: the automated driving system (ADS). However, little effort has thus far been spent on understanding the implications of changing from a human driver to an ADS for the physical road infrastructure and vice versa [11]. ADSs are expected to contribute to safer roads. To ensure this outcome, the design and maintenance of road infrastructure must be correctly interpreted by both humans and machines. To evaluate existing quality parameters such as the visibility of road markings considering automated driving systems can help future-proof road infrastructure. 

Lane detection is considered to be important for any ADS [12,13,14,15] and is dependent on the visibility and consistency of lane markings [16,17]. Identification of lanes via longitudinal road markings is usually done with cameras, in either monocular or stereo vision [13,14,18]. Lane detection systems need to overcome several challenges, including knowing which lane a vehicle is in on a multilane road, separating road markings from other longitudinal lines such as asphalt surface cracks and guardrails, and accurately detecting worn markings, even in challenging light and weather conditions [12]. Worn markings pose a similar safety concern for human drivers [19,20]. 

Given the continuing development of vehicle technology, ADAS is a promising source of data on both the condition of road infrastructure elements and how to facilitate automated detection of such elements. This data is valuable input for road asset management systems and can facilitate safer roads through design and management that play to the strengths of driver support systems. The aim of this study was to determine whether conventional methods of assessing road marking quality through retroreflectivity are consistent with lane marking detection systems found in cars that typically rely on cameras. The conventional method utilizes a retroreflectometer, and those results were then compared to results from a lane departure warning system. A similar comparison was made by [18]. In their study, they used two different vehicles to cover a 100 km stretch of highway outside Prague, one with a lane departure warning system and another with a retroreflectometer attached. In contrast, this study used a mobile retroreflectometer on the same vehicle that had the lane departure warning system. In this way, the data collected were known to be consistent for the two sources. Moreover, this study investigated the differences between freeways and county roads, as well as both daytime and night-time conditions. Another effort to relate the quality of road markings to the success of automated camera-based lane detection was presented in [19]. The research compared the retroreflection measured by a roof-mounted Automatic Road Analyzer to the results of a Mobileye lane detection system. The Mobileye system was mounted on the vehicle equipped with the Automatic Road Analyzer and results suggest that a Q_d_ higher than 135 mcd/m2/lux improves the detection of lane markings using a Mobileye lane detection system. A reference test system for camera-based lane detection was developed in [15]. In this research, it is stated that there are no available standards or benchmarks to assess the quality of either road markings or perception algorithms associated with these. While it could be argued that there are benchmarks for road markings, e.g., there are both European [20] and Nordic standards for evaluating the quality of road markings [5], it highlights the need for similar standards for machine-vision applications. Where the work in [15] compares videos annotated with additional data by mobile retroreflectometer, the video and reference data were not gathered simultaneously. There is no information on the retroreflectometer readings in terms of how it logged data (time-based versus distance-based) or whether the readings were averaged over some time or distance. Due to these two reasons, there are unknown sources of errors in the use of the mobile retroreflectometer as a reference. While the study in [15] investigates different lane detection algorithms used on annotated videos, the research described in this paper shows how the output from the LDW system could be used directly by road owners and operators (ROOs). This is an important difference as the video on which the LDW system relies is generally not available while the output of the LDW system is a binary output that could easily be shared between vehicles and ROOs.

Two of the most commonly used properties to assess the quality of road markings are the luminance factor under diffuse illumination, Q_d_, and the coefficient of retroreflected luminance, R_L_. Q_d_ is a measure for visibility under daylight conditions, during which natural light hits the marking and is dispersed in all directions. R_L_ is used under night-time or otherwise dark conditions, during which an active light source is directed toward the marking and reflection is measured. Mobile retroreflectometers have been developed that can measure the night-time retroreflectivity of road markings using a laser beam to simulate vehicle headlights, and these measurements are independent of ambient light levels. Retroreflectometers are designed to match the entrance and observation angles from a driver’s eye to the road marking [21]. The coefficient of retroreflected luminance, R_L_, measured in millicandelas per lux per square meter (mcd/m^2^/lux), is defined by the American Society for Testing and Materials as the ratio of the luminance of a projected surface to the normal illuminance at the surface on a plane normal to the incident light [22]. 

Today, it is common for road agencies to use reflectorized pavement markings that contain glass or ceramic beads. Nonreflectorized markings also have reflectivity based on the type of material, but reflectorized markings are generally preferred because of their higher reflectivity at night [23]. Experts have highlighted the need for a minimum requirement for retroreflectivity [24]. In Europe, a minimum retroreflectivity of 150 (mcd/lux/m^2^) has been suggested for dry conditions, which is in line with the minimum requirement already in effect in several European countries [10]. In the US, a supplemental notice of proposed amendment (SNPA) establishes a revised set of standards to be incorporated in the American Manual on Uniform Traffic Control Devices [25]. The SNPA suggests a minimum retroreflectivity level of 50 mcd/lux/m^2^ for speed limits of greater than 35 mph. For speed limits above 70 mph, the minimum retroreflectivity level suggested is 100 mcd/lux/m^2^.

Contrast has been identified as important for humans to be able to detect road markings [26,27] and stay in their lanes [28]. For ADSs, the need for contrast has likewise been identified [8,19,29,30].

Retroreflectometers, like the equipment used in this study, provide only R_L_ and contrast values. These were also crucial in the study by Lundkvist and Fors [31] to determine whether current Swedish requirements for road markings were sufficient for the LDW system of a Volvo S80. They used a mobile retroreflectometer to obtain the retroreflective value of the marking, R_L dry_, and contrast, along with an optocator to provide macrotexture readings of the road surface. Tests were performed on roads of different classes and under varying conditions (wet, dry, day, and night). Their study showed that the LDW system worked well on primary roads during daytime conditions (wet and dry), with a detection rate of about 99%. While the same was true for night-time dry conditions, for night-time wet conditions the rate dropped to 92%. For secondary roads, the detection rate for daytime was 80%, and the lower detection rate was accredited to worn or dirty markings. Lundkvist and Fors stated that the detection rate for secondary roads during night-time wet conditions was very low on some roads, and no overall rate of detection for this case was given. The reason for the lack of detection was thought to be the low contrast between the markings and pavement as well as the fact that the secondary roads did not have retroreflective markings. Another case that proved difficult for the LDW system was when the sun was low on the horizon, which caused detection rates as low as 50%. The study identified lower limits for R_L_, Q_d,_ and their respective contrasts. The contrast for Q_d_ was calculated by using Equation (1) and equivalently calculated for C_RL_. The lower limits are presented in Table 1.
(1)$$C_{QD}=\left|\frac{Q_{marking}-Q_{road\;surface}}{Q_{road\;surface}}\right|$$

The visibility of markings is affected by the available ambient light. Night-time light conditions provide lower visibility for both humans and machines [8,21], although some machine vision systems have shown better results during wet night-time conditions than during wet daytime conditions [29]. Bright illumination in low light conditions, for instance from street lights or headlights, can saturate the image and make the detection of lane markers challenging, especially if the road marking is aged and worn [32].

Lin et al. [33] proposed edge smoothness as a new quality indicator specifically for machine vision systems. Algorithms detect road markings in images by using lines, edges, and rectangular shapes; therefore, their research has suggested that road markings with smooth edges would be easier to identify, as they resemble the straight lines used in the algorithms’ computations. Edge smoothness has not traditionally been measured in the field: to do so, both the measurement device and relevant thresholds would need to be determined. Edge smoothness was not evaluated in this study, as relevant data for this type of analysis were not available from the equipment used.

Finally, speed impacts the ability to identify markings, as speed determines the amount of time the driver has to detect markings. Zwahlen and Schnell [26] performed numerous simulator studies using human drivers. Their studies showed that an increase in vehicle speed required a considerable increase in minimum retroreflectivity levels to attain the same preview time. For a machine vision system, higher speeds would mean fewer frames for analysis and less time for processing.

This paper investigates if ADAS functionality, represented by the case of LDW, provides an efficient way to monitor the road infrastructure and assesses how traditional measures of quality, for example, retroreflection, affect the outcome of lane detection by the LDW. The following section describes the design of the experiment and the data collected. Section 3 describes the results of the data analyses, while the implications and future research avenues are discussed in the fourth and final section.

## 2. Methods

In May 2019, a test was performed by the paper’s authors to compare the measurements of a retroreflectometer with the lane detection results from a car with LDW functionality. A car with a built-in lane detection camera was outfitted with a retroreflectometer and driven on three routes between the locations of Oslo, Moss, and Drøbak in Norway (Figure 1). Route 1 was on freeways (E-road 6), and routes 2 (daytime driving on county roads 152, 155, 156, and 1386) and 3 (night-time driving on county roads 51, 60, 152, 316, and 1422) were on county roads. The higher maintenance level of the freeway meant that those road markings were subject to stricter requirements, although this does not always guarantee that freeways have higher levels of retroreflectivity and contrast [34]. 

The requirements for retroreflection from the Norwegian road manual are dependent on winter maintenance classes, which are divided into five levels. Freeways require the highest level of winter maintenance, class DkA, while county roads call for the second-highest level, the DkB class. For both the DkA and DkB classes, the minimum required RL is dependent on the average annual daily traffic (AADT). If AADT is below 5000, as seen along parts of county roads, the minimum R_L_ is 100 mcd/m^2^/lux. For AADTs greater than 5000, which includes the freeways and remaining sections of county roads, the requirement is 150 mcd/m^2^/lux [35].

The conditions during the test were dry asphalt, and data were recorded for both daytime and night-time. In the freeway case, the data were recorded along the same route for both day and night (route 1 in Figure 1). For the county roads, the data captured for day and night did not follow the same route, as a malfunction in the retroreflectometer caused by low temperatures at night prevented the logging of data. For the county road, daylight data were from route 2, and night-time data were from route 3 (Figure 1). A visual comparison of routes 2 and 3 indicated that there were no large differences in the standards of the roads or road markings. 

The daytime measurements were taken on a generally overcast day, with the car reporting a lux value for ambient light of around 10,000. For reference, bright sunlight produces lux values of between 50,000 to 100,000 [36]. The night-time measurements were performed from approximately 9:30 pm to midnight, with most lux readings taken between 0 and 11.

The test was performed by using a single 2018-model car with an LDW system. The data from the car were made available from the manufacturer. The LDW system produced data in four discrete values, indicating whether the system registered no detection, left-hand detection, right-hand detection, or detection on both sides. The LDW system used a mono camera mounted behind the rear-view mirror, a height similar to the eye height of humans. It also used data from an odometer and an inertial measurement unit; the latter indicated the forces acting on the car and its heading. 

A Laserlux G7 retroreflectometer from RoadVista was used to register the retroreflection and contrast of the longitudinal road marking. It was attached to the car with the LDW system as shown in Figure 2. The retroreflectometer measured the quality of road markings while driving at highway speeds, registering data averaged over 30 m. The distance was set to 30 m as this setting gave a consistent flow of registered data; smaller values often resulted in improperly recorded data. A secondary reference camera was installed within the vehicle to capture video logs of the test conditions.

The Laserlux retroreflectometer can only measure road markings on one side of the car at a time. Zwahlen and Schnell [37] found in their research that the distance at which humans can see road markings is mainly governed by the visibility of the right edge line in the case of a fully marked road. The retroreflectometer was therefore attached to the right side of the car with respect to the driving direction, and the car was driven in the rightmost lane throughout the test, thus detecting the fog line. The retroreflectometer’s laser beams hit the road at approximately 1.5 m in front of the car on the fog line (Figure 2). 

The laser beams had an impact width of about three times the width of the freeway markings and about five times the width of the county road markings (because of the difference in road marking widths). The data were collected by driving so that the retroreflectometer’s lasers hit the fog line; given the impact width, the laser was therefore thought to have hit the markings throughout most of the experiment. Some discrepancies were expected, as the car was driven manually. 

The differences in lane marking detection by a camera and a retroreflectometer are worth noting. In comparison to retroreflectometers, a car’s camera has a relatively high position, which provides a much greater vision angle and viewing distance (Figure 2). In addition to the different positions of the sensing media, the camera is more dependent on ambient light than the retroreflectometer. This is because the latter has active lighting in the form of laser beams, which hit the markings so their reflection can be measured, making the system insensitive to ambient light. Because of these differences, it was of interest to determine whether detection by the retroreflectometer, an established quality assessment tool for road markings, would be comparable to detection by a typical lane departure warning system. Table 2 shows the data collected. 

The mobile retroreflectometer reported contrast values based on Equation (2): (2)$$C=1-\frac{return\;signal\;\left(pavement\right)}{return\;signal\;\left(marking\right)}$$

This value can be converted to a contrast ratio by Equation (3): (3)$$CR=\;\frac{-1}{C-1}$$

The retroreflectometer collects data based on distance, whereas the car’s data capture is based on time. These differences resulted in different data densities for different speeds. The car had a high data sample rate, 12.5200 Hz (Table 2). The retroreflectometer collected data and averaged them over 30 m. This provided low data densities at low speed and denser data at higher speeds. 

The four previously described categories of lane detection were converted to a binary set in which values for *None* and *Left-hand* detection were set at 0, and values for *Both* and *Right-hand* detection were set at 1 to identify when the right-side marking was detected. The other data were resampled with interpolation to match the sample rate of the *Lane detection* values by using the Python data analysis library *Pandas*. The reason for choosing *Lane detection* values as the sampling reference was their binary nature. Resampling the lane detection values would change them from discrete to an artificial continuous set of values and make the analyses less meaningful. The data were time-series data which are classified as ordered data. They could therefore be merged on the time variables using Pandas function *pdmerge_ordered* with forward fill. Forward fill propagates the last valid observation forward in the case of missing values.

Binary logistic regressions were performed in SPSS to investigate whether the data from the retroreflectometer could predict the outcome of the lane detection. Binary regression was used since the predicted outcome, whether the LDW function detected the road marking, was a binary outcome. The binary logistic regression is expressed as the estimated probability that Y equals 1 given input X, where Y ε {0,1}:Prob{Y = 1|X} = [1 + exp(−Xβ)]^−1^(4)

The regression parameters β are estimated by the method of maximum likelihood [38]. Four cases were considered: two sets of data from the freeway between Oslo and Moss (route 1), one for daytime, and another for night-time. Another two sets were from the county roads. The day case covered roads between Drøbak and Oslo (route 2, Figure 1), and the night case involved roads between Moss and Oslo (route 3, Figure 1). Regarding the binary logistic regression, the predictors were *Retroreflection, Contrast, Vehicle speed,* and *Ambient light*, and the dependent variable was *Lane detection*. *Retroreflection* was chosen as a predictor since it is the most common measure of road marking quality and *Contrast* was chosen due to the body of research indicating its importance for both human and machine perception of lanes. The *Vehicle speed* was selected as a predictor because the retroreflectometer logged data based on distance while the vehicle logged based on time, and *Ambient light* was chosen as visibility is related to available light.

## 3. Results

The binary logistic regression model determined to what extent the *Retroreflection*, *Contrast*, *Vehicle speed,* and *Ambient light* could be used to predict whether the LDW system in the vehicle detected the fog line. 

Examples of daytime situations are shown in Figure 3. Please note that these are from the reference camera and not from the car’s camera.

Figure 4 shows images from the freeway (left) and county road (right), both with and without street lighting, also taken from the reference camera. The retroreflectometer’s lasers is visible in the images in Figure 4 and, as previously stated, was not dependent on ambient light (unlike the camera). In the lower right image, the headlights saturate the image to the extent that the road marking is difficult to discern from the pavement, highlighting the challenges of camera detection with low ambient light. 

### 3.1. Binary Regression for Freeway and County Road in Daytime and Night-Time Conditions

The results of the binary regression analysis are presented in Table 3. The cut value was set to 0.5 for the analysis in Table 3, meaning that if the probability of lane marking being detected was greater than 50%, it would be classified as a positive detection (=1).

In the table, 0 indicates that no lane marking was detected, while 1 indicates that the road marking was found. Table 3 shows that the outcome of the lane detection’s functionality was correctly predicted by the model in 92.1% of cases for the freeway and 92.8% of cases for county roads in the daytime. Regarding night-time, the results were 93.1% for the freeway and 98.1% for county roads. There was a higher accuracy level for no detection of lane markings on the freeway in both daytime and night-time. The same was true for the county roads at night-time, although the difference was not as distinct. With respect to the county road daytime case, the accuracy was similar between no marking detected and a marking found. There was a considerable difference between the freeway and county roads in the share of cases concerning where markings were detected versus where they were not. The county roads had higher rates of unmarked roads than the freeway. This was expected because of the county roads’ lower maintenance level. 

### 3.2. Significance of the Predictor Variables

In the case of binary logistic regression, it is assumed that observations are independent and that the explanatory variables are not linear combinations of each other. This ensures that multicollinearity is not introduced into the analysis. In this model, the two predictor variables *Retroreflection* and *Contrast* were related by Equation (2). According to Midi et al. [39], “Multicollinearity does not reduce the predictive power or reliability of the model as a whole; it only affects calculations regarding individual predictors.” To investigate the impacts of the four individual predictors without multicollinearity issues, the binary regression analyses were performed using the *Retroreflection* and *Contrast* predictors separately, keeping the other two predictor variables. As the values for *Vehicle speed* and *Ambient light* remained almost identical in the analyses with *Retroreflection* and *Contrast,* respectively, their values were averaged, as shown in Table 4. In Table 4, B represents the regression weights, that is, the βs from Equation (4), S.E. is an abbreviation for Standard Error, Sig. is an abbreviation for statistical significance, and the Exp(B) is the exponential of B also known as the odds ratio which signifies how the odds change with respect to changes in the associated predictor variable.

In all cases, *Ambient light* had no impact on a successful prediction, as indicated by the exponential(B) or odds ratio ≈ 1 in Table 4. It is possible that the scenarios used in this study did not provide a wide enough range of ambient light values to correctly identify its significance in camera-based lane detection. However, the results were in line with Table 2, which showed that the best prediction of road marking detection occurred at night, suggesting that the LDW system was not dependent on ambient light. These results indicate that the combination of low ambient light and headlights, as shown in Figure 4, does not pose a problem for machine vision lane detection. In fact, the model performed overall better at night, which was contrary to findings by Lundkvist and Fors [31] and Borkar et al. [32].

Similarly, regarding *Retroreflection*, the odds ratio was close to 1.0 in all four cases, meaning it had very little effect on predicting a correct outcome. As retroreflection is the most common indicator of quality in traditional road marking evaluation, this is of interest and suggests that other parameters (e.g., contrast and edge smoothness) might be needed to evaluate the quality of road markings for machine vision-based systems.

*Vehicle speed* proved to be the most influential predictor of the analysis. That was not surprising, as the data collection used as input was directly dependent on speed. When a vehicle travels faster, more distance is covered, providing more data points from the input data generated by the retroreflectometer. This means that the faster the vehicle with the retroreflectometer travels, the more data points the retroreflectometer has on which to base its prediction of the LDW system’s outcome, relative to the time constant machine vision. In both daytime conditions, the analyses yielded an odds ratio of approximately 3, meaning that a one-point increase in speed would produce a threefold increase in successful predictions. Under night-time conditions, the results showed a clear difference between road types. On the freeway, the effect of speed was lower, at an odds ratio of about 1.5, whereas on the county road, the odds ratio for speed was about seven times higher than for day, at 22.1. To interpret these results, the difference between road types must be considered. For instance, vehicle speeds on the freeway had a smaller range and higher mean, as driving at the speed limit on the freeway resulted in a consistent speed of about 80 km/h. On the county roads, speeds varied more, from low speeds in corners and roundabouts to about 60 km/h in straight sections. The freeway case, therefore, produced a higher and more consistent rate of data from the retroreflectometer, the input data. Regarding the county roads, the range in speed meant that at low speeds the input data were very scarce in comparison to the output they were meant to predict, while at higher speeds the input data were denser than the time constant LDW data. 

To further understand the great difference in odds ratios for the *Vehicle speed* on county roads given the time of day, an overlapping section on route 2 (daytime) and route 3 (night-time) was isolated. The overlapping segment is shown in Figure 5.

In this section, no positive identifications of lane marking could be found in the night-time case, yet there are numerous positive identifications for the daytime case (Figure 6). 

The data were collected on consecutive days and under the same weather conditions, which means that the main difference was the amount of ambient light available. The videos from the stretch of road were reviewed to find causes for the lack of detections. Some on-coming traffic was noted, which can be problematic for the LDW [31], but this would not have accounted for the complete lack of detections at night. The type of marking used, that is, paint or thermoplastic, was rudimentarily checked by manually comparing the stretches of road for the day and night scenario, respectively, where information on what material was used was available. The night route had more spray thermoplastics (1–1.5 mm thickness) than the day route, which had almost entirely preformed thermoplastics (3 mm). This could explain the overall trend of much fewer successful lane detection for night versus day. For the overlapping segment, however, the marking was thermoplastic marking in both cases. The marking used on county road was 2 mm thick thermoplastic compared to the 3mm thick thermoplastic used on the freeway. In addition to thickness, application and wear can factor into the marking’s visibility. As ambient light was shown not to be a predictor for the outcome of the LDW, the material and thickness used for road marking could be investigated as a predictor for automated lane detection in future studies. 

*Contrast* was the second most influential predictor, contributing in the range of 1.64 to 1.71 for the different scenarios, with slightly higher values for daytime conditions. This means a unit change in contrast would result in 1.64–1.71 times the odds of getting a correct prediction. Lundkvist and Fors [31] found that the successful detection of road markings by an LDW system in daytime did not require high contrast, and the results of binary logistic regression indicated that contrast was a better indicator of whether the LDW system detected markings than the more widely used Q_d_ or R_L_. *Contrast* has been identified as an important parameter for both human and machine detection [8,34], a finding supported by these results. The level to which it contributed was modest but consistent across all scenarios. 

### 3.3. Evaluation of Possible Threshold Values 

Receiver operating characteristic (ROC) curves can be used to look for threshold values for successful detection. The ROC curve summarizes the trade-off between the true positive rate and false-positive rate for a predictive model and is used as a diagnostic tool for the model. The four predictor parameters were used as test variables, and the positive lane detection was set as the state variable. Figure 6 shows the ROC curves for the four cases, with sensitivity plotted on the *y*-axis and 1-specificity on the *x*-axis, as per convention. The sensitivity and specificity are defined by Equations (5) and (6) [40]:(5)$$Sensitivity=\;\frac{True\;positives}{True\;positives+\;False\;negatives}$$
(6)$$Specificity=\;\frac{True\;negatives}{True\;negatives+\;False\;positives}$$

In the figure, the larger the area under the curve (AUC) for a given line, the more accurate the variable is at predicting the outcome. Threshold values for the variables are located at points as far upward and to the left on the curve being analyzed as possible. The higher up on the *y*-axis, the more true positives are included. However, as the x values increases, so does the rate of false positives.

A curve close to the diagonal reference line indicates that the test variable is not useful for distinguishing between a positive or negative outcome. In Figure 7, ambient light lies close to the diagonal in all cases except for the freeway night-time case, which has a slight distinction. The ROC curves indicate that ambient light is not a good indicator of whether the lane is detected, as was seen in the regression analyses.

Figure 6 shows that with respect to daytime driving, *Retroreflection* and *Contrast* also seemed to have little predictive quality. Under night-time conditions, the AUC is larger for these two variables, and within these two cases, the curves for the county road are the best in terms of establishing a threshold cut-off. Resulting in a high level of sensitivity and relatively low value for 1-specificity. 

In the tables that accompanied the ROC curves, the value of the test variable that corresponded to an identified threshold point on the curve (high y-coordinate, low x-coordinate) could be extracted. These threshold values are indicated in Figure 7.

The ROC curves for *Vehicle speed* have large AUCs, indicating that a high level of sensitivity and specificity can be obtained without introducing too many false positives or negatives. As with the binary logistic regression analysis, they have the largest impact on predicting the outcome. Additionally, Figure 7 shows the chosen threshold points and values for the test variables. With respect to *Vehicle speed*, the threshold values lie between 56.14 and 57.02 km/h.

The night-time contrasts thresholds reported were converted to contrast ratios by Equation (3), and county roads were associated with a threshold contrast ratio of 8, whereas for the freeway the threshold equated to a contrast ratio of 2.74. The required contrast for detection of road markings by LDW identified by Lundkvist and Fors (2010), C_RL_, was 3 (Table 1). Unfortunately, as the retroreflectometer used was not the same, and data were not derived in the same manner, it is difficult to compare the results of the two studies. Substituting Equation (2) for C in Equation (3) would give Equation (7) for determining CR within this study, while Lundkvist and Fors used Equation (8). The equipment used in this work did not report R, therefore, it was not possible to calculate the contrast in the same way as in Equation (8).
(7)$$CR=\frac{return\;signal_{marking}}{return\;signal_{pavement}}$$
(8)$$C_{RL}=\left|\frac{R_{marking}-R_{road\;surface}}{R_{road\;surface}}\right|$$

## 4. Discussion

This study compared data from a conventional performance measure for longitudinal road marking, a retroreflectometer, to those from a modern ADAS feature, LDW, which utilizes a camera within the vehicle. The aim was to determine to what extent the conventional retroreflectometer could predict the outcome of the LDW’s ability to detect lane markings. The results give insight into both whether the established quality parameters for road markings are suitable for automated detection systems and whether crowdsourced data from vehicles can be used to monitor pavement marking conditions. 

The binary regression analyses showed that the model, correctly predicts the result of the car’s lane detection in 92.1% to 98.1% of cases, depending on driving conditions. 

The results indicate that the in-vehicle camera’s ability to detect lane markings could potentially be used as a surrogate for conventional methods of assessing the status of pavement markings. The impact of the specific input variables on the model was also considered to better understand how traditional metrics associated with the assessment of lane markings impact the LDW predictions.

*Ambient light* and *Retroreflection* were found to have minimal impact on the success of predictions. The lack of impact of ambient light might be due to the limited range of input values, as the measurements were taken in either high or very low ambient light conditions and did not cover the middle ground. Another possibility is that the LDW system is not dependent on ambient light and that headlights provide sufficient lighting to detect markings. Additional investigations are likely needed, but these results indicate that the LDW system is not dependent on street lighting which is a useful input towards facilitation for driving automation in road design.

Retroreflection is the most common parameter for the quality assessment of road markings today. This work indicates that the retroreflection value in itself is not important for the successful identification of road marking by machine vision. The algorithms used to detect markings search for lines, edges, and rectangular shapes, making the contrast between the road surface and road marking more critical than solely the amount of light reflected from the marking. Threshold values for *Contrast* could only be suggested in nighttime driving, and for these limited tests, the contrast ratio thresholds were found to be 2.74 for the freeway and 8 for the county roads. Contrast, as well as edge smoothness, are suggested as quality parameters for machine vision detection of lane markings. Yet, as contrast is calculated differently depending on the method used, it may be difficult to specify a universal threshold value.

The analysis showed that *Vehicle speed* had the strongest impact on the success of predictions. This is because of its direct relation to the amount of data captured. The speed of the vehicle carrying the retroreflectometer should be 57 km/h or higher for best results.

The high probability of predicting the outcome of the lane detection feature, despite relying on few input variables (*Retroreflectivity* and the derived *Contrast*) suggests that the LDW system is more dependent on longitudinal road markings than other elements of the road geometry such as the road’s width, edge, or centerline. 

Further investigations should be performed to affirm the indication that LDW functionality can be used to monitor the state of road markings. This could include using different makes of cars and collecting data from more roads. The indication that ambient light does not affect the success of the LDW system could be researched further by driving in different conditions of natural and artificial light. The effect of road marking material and thickness on lane detection by the LDW is encouraged as it could be particularly useful for adapting design and maintenance of roads to support automated driving features. Harmonization of measurement techniques would be beneficial in supporting earlier findings. For instance, the contrast measurement in [31] is not directly comparable to the contrast measurement in this paper. Likewise, the retroreflectivity in [19] is of Q_d_ which is dependent on ambient light, while this paper used an active laser measuring R_L_.

This research has shown that ADAS functionality such as LDW provides an efficient way to monitor the road infrastructure. The work identified that contrast between the road marking and the adjacent road surface is of more importance than the retroreflectivity of the road marking material, suggesting that contrast should be a metric for quality of road marking to start including automated driving functions in the design and maintenance of roads. Gaining knowledge about how well infrastructure designed for humans also supports automated driving systems will be useful for future-proofing roads. It will help reap the promised safety benefits of automated driving features, from today’s ADAS to more comprehensive future ADS. Additionally, crowdsourcing is an efficient way of monitoring the state of road infrastructure and can provide systems for predicting road maintenance, such as the PMMS, with highly valuable data. Understanding how infrastructure is “seen” by both humans and ADSs will allow engineers to prepare for higher levels of automation by taking incremental steps in which road infrastructure design and maintenance can be developed in parallel with vehicle automation.

## Figures and Tables

**Figure 1 sensors-21-01737-f001:**
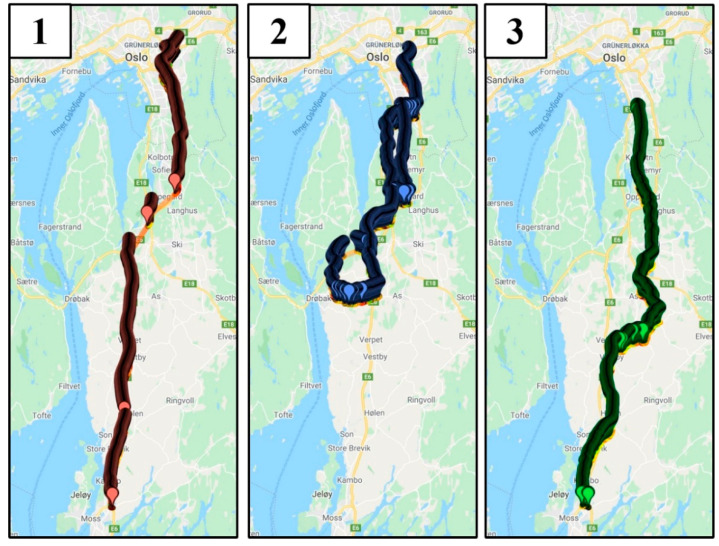
Routes driven: (**1**) freeway route for both day and night cases, (**2**) county roads daytime route, (**3**) county road night-time route. Visualization: http://geojsonviewer.nsspot.net/ (accessed on 20 February 2021).

**Figure 2 sensors-21-01737-f002:**
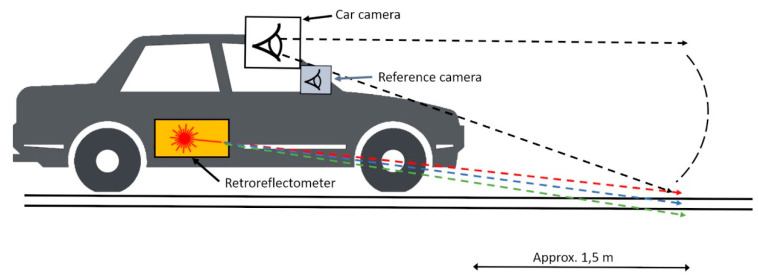
Illustration of the test set-up.

**Figure 3 sensors-21-01737-f003:**
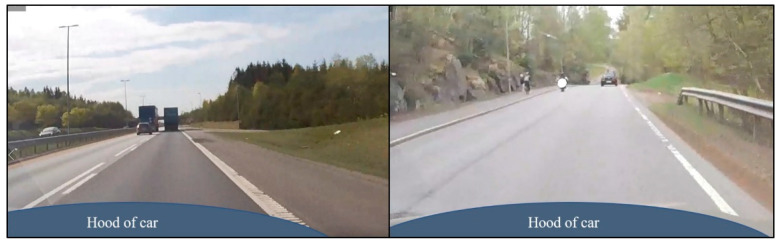
Light conditions for daytime driving for freeway (**left**) and county road (**right**).

**Figure 4 sensors-21-01737-f004:**
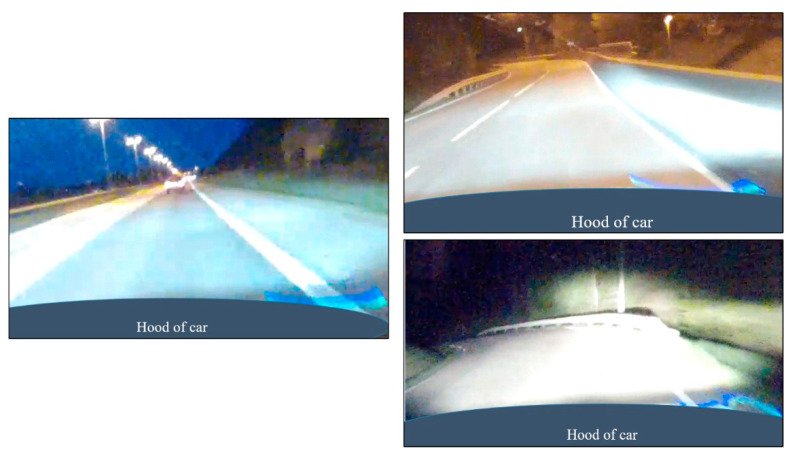
Light conditions for night-time driving for freeway (**left**), county road with street lighting (**upper right**), and county road without street lighting (**lower right**).

**Figure 5 sensors-21-01737-f005:**
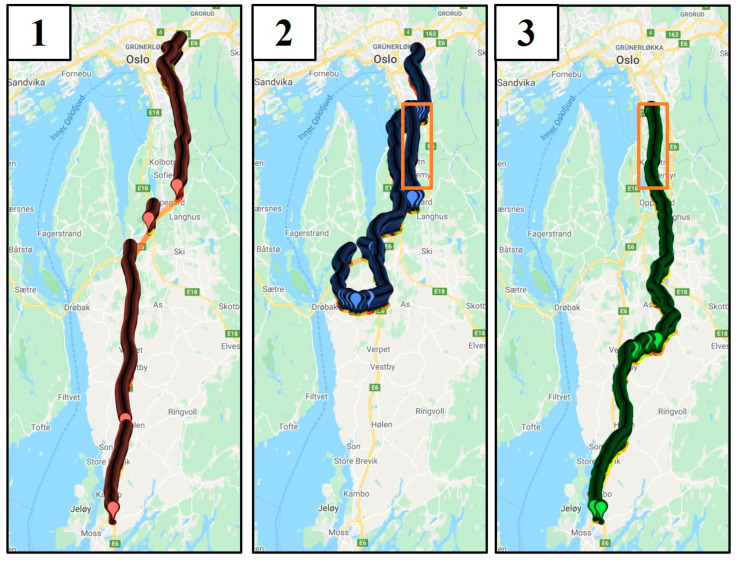
Overlapping road sections on county roads for daytime (2) and night-time (3) driving.

**Figure 6 sensors-21-01737-f006:**
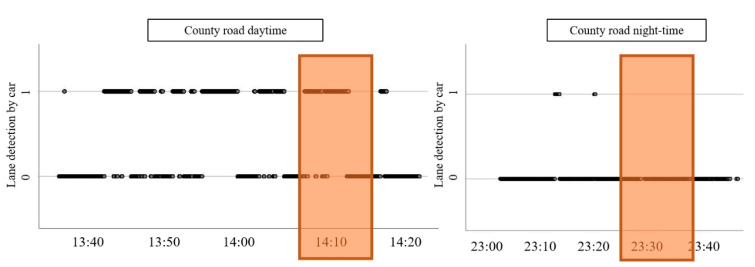
Overlapping segment on county roads for daytime vs. night-time.

**Figure 7 sensors-21-01737-f007:**
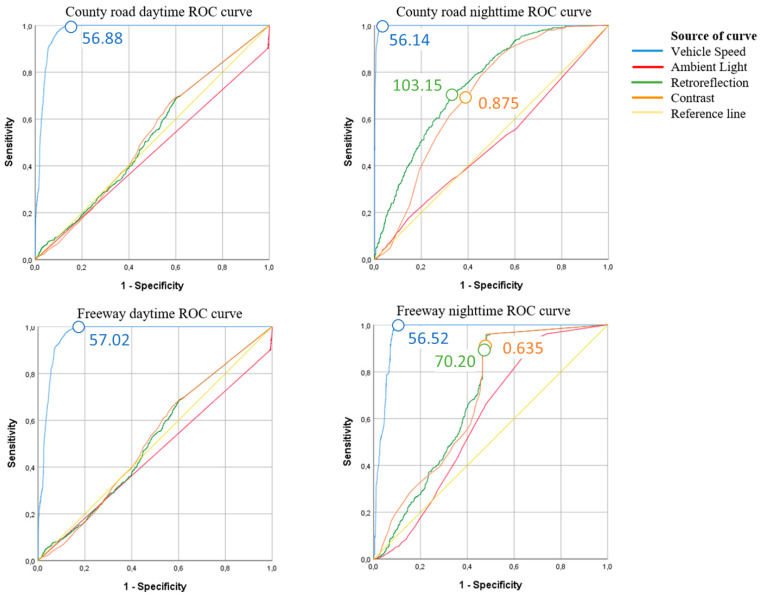
Receiver operating characteristics curves and suggested threshold values for the predictor variables.

**Table 1 sensors-21-01737-t001:** Estimated lower limits for detection by LDW [31].

Condition	Lowest Q_d_ [mcd/m^2^/lux]	Lowest R_L_ [mcd/m^2^/lux]	Lowest C_QD_	Lowest C_RL_
Daytime, dry	≈65		≈0.08	
Daytime, wet	≈65		≈0.08	
Night-time, dry		≈70		≈3.7
Night-time, wet		≈20		≈3.0

**Table 2 sensors-21-01737-t002:** Data used for analysis.

Retroreflectometer Laserlux G7			Car with Lane Departure Warning		
Data	Sample rate	Unit	Data	Sample rate	Unit
*Latitude, longitude*	30 m	WGS84	*Latitude, longitude*	12.5Hz	WGS84
*Vehicle speed*	30 m	km/h	*Vehicle speed*	200 Hz	m/s
*Retroreflection average*	30 m	mcd/lux/m^2^	*Ambient light*	100 Hz	lux
*Contrast*	30 m	Contrast (Equation (1))	*Lane detection*	50 Hz	Yes/No

**Table 3 sensors-21-01737-t003:** Classification of all cases: freeway and county roads in daytime and nighttime.

Freeway Daytime (Lux Values: Mean = 9663, Median = 10,000), (Cut Value is 0.5)					Freeway Night-Time (Lux Values: Mean = 10, Median = 7), (Cut Value is 0.5)				
Observed		Predicted			Observed		Predicted		
		Lane detection		Correct (%)			Lane detection		Correct (%)
		0	1				0	1	
Lane detection by car	0	959,887	55,543	94.5	Lane detection by car	0	369,022	28,771	92.8
	1	60,635	401,120	86.9		1	35,314	461,956	92.9
Overall (%)				92.1	Overall (%)				92.8
**County road daytime (lux values: mean = 9,726, median = 10,000), (cut value is 0.5)**					**County road night-time (lux values: mean = 6, median = 3), (cut value is 0.5)**				
Observed		Predicted			Observed		Predicted		
		Lane detection		Correct (%)			Lane detection		Correct (%)
		0	1				0	1	
Lane detection by car	0	1,256,579	55,058	95.8	Lane detection by car	0	1,064,740	17,027	98.4
	1	67,802	401,812	85.6		1	7,020	202,021	96.6
Overall (%)				93.1	Overall (%)				98.1

**Table 4 sensors-21-01737-t004:** Significance of predictor variables.

County Road Daytime					County Road Nighttime				
	B	S.E	Sig.	Exp(B)		B	S.E.	Sig.	Exp(B)
Retroreflection	0.003	0.000	0.000	1.003	Retroreflection	−0.010	0.012	0.000	0.990
Contrast	0.538	0.000	0.000	1.712	Contrast	0.514	0.067	0.000	1.672
Vehicle Speed	1.127	0.002	0.000	3.087	Vehicle Speed	3.097	0.012	0.000	22.127
Ambient Light	0.000	0.000	0.000	1.000	Ambient Light	0.003	0.001	0.000	1.003
Constant	−19.30	0.043	0.000	0.000	Constant	−50.89	0.207	0.000	0.000
**Freeway daytime**					**Freeway nighttime**				
	**B**	**S.E.**	**Sig.**	**Exp(B)**		**B**	**S.E.**	**Sig.**	**Exp(B)**
Retroreflection	0.003	0.000	0.000	1.003	Retroreflection	0.004	0.000	0.000	1.004
Contrast	0.538	0.008	0.000	1.713	Contrast	0.495	0.017	0.000	1.640
Vehicle Speed	1.076	0.002	0.000	2.932	Vehicle Speed	0.429	0.001	0.000	1.536
Ambient Light	0.000	0.000	0.000	1.000	Ambient Light	−0.011	0.000	0.000	0.989
Constant	−18.39	0.043	0.000	0.000	Constant	−8.100	0.023	0.000	0.000

## Data Availability

The data presented in this study are available at https://github.com/ResearchAne/DataAnalysesLDW (accessed on 5 February 2021).

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
