# Peer review of "Using ADAS to Future-Proof Roads—Comparison of Fog Line Detection from an In-Vehicle Camera and Mobile Retroreflectometer"

_sensors, 2021, doi:10.3390/s21051737_

Round 1

Reviewer 1 Report

Can you describe better the main contributions?

How did you define the RTT algorithm?

You should define better the safety constraints.

Why not used the spline to define better path planning?

Fig. 1 needs to be better described and draw

How do you define the best DNN architecture?

Author Response

Thank you four reviewing our paper, please see the attachment for revisions made.

Reviewer 2 Report

N/A

Author Response

Thank you four reviewing our paper.

Reviewer 3 Report

he authors present an interesting comparison to evaluate the effectiveness of an in-vehicle camera in detecting road fog lines and estimate threshold values of road markings characteristics. They propose a binary logistic regression model for road marking detection using data collected by a retroreflectometer mounted on the right side of a car equipped with ADAS. The paper is well written and structured. However, some major revisions are required before being considered for publication.

Detailed comments:
- The structure of the paper is missing at the end of the introduction. Please revise accordingly.
- The authors used the Python Pandas library for data interpolation without specifying which interpolation technique was considered. Please revise accordingly.
- I suggest adding the binary logistic regression model formulation for completeness and better understanding.
- It seems that the authors used SPSS software for the binary logistic regression, but it should be specified in the text and better explained all the column headings in Table 4 since not all readers use the same software for regression analysis.
- According to Table 4, it is strange to observe that, for example, the ambient light variable has no impact on prediction (B=0) but its p-value (Sig.) is always 0 (high significance). Please explain in the paper or check the correctness of values reported in the table.
- I suggest completing Fig. 5 adding the corresponding road segment on the map.
- Adding some further developments could complete the final discussion.

Author Response

(The authors gave the same response as above.)

Reviewer 4 Report

The paper deals with prediction the the outcome of the lane detection algorithm (ADAS implemented in vehicle) based on data obtained by retroreflectometer.  Mine main concerns regarding paper:

Abstract does not put strong impression on the reader. It is not concise as it should be. I suggest authors to revisit it (try to shorten it to make it more concise and direct).

Introduction is too long. The authors are giving too much information (which is maybe not so important for introduction) and the important ones are missing (what is the current state of the art, what is concise contribution of the paper). E.g. some details regarding retroreflectometers can be moved to appropriate section (maybe new section 2).

Contribution of the paper should be clearly pointed out in the end of Introduction. 

The related work section is missing. Some of the papers are mentioned in Introduction, but I found this rather weak.

I found references somewhat old. There are several more recent references which should be introduced in paper and summarized (e.g. [1-2]). The authors are suggested to make more detailed investigation of the current state of research in the field of ADAS and their dependence on road markings (especially lane departure warning systems)

Method section has lot of deficiencies. it is not clear why authors used logistic regression from the whole universe of ML classifiers.. Also, how the input variables were selected?

Result section is missing explanation regarding data divison. If the all data is used to estimate coefficients of logistic regression then I found this quite wrong since authors would like to show to what extent such model can predict lane detection (so unseen data should be used for testing)

Some parts can be merged. For example I don't see any value in reporting classifier accuracy at 0.5 and then later AUC is reported which basically covers different thresholds.
Also there is a lot of explantion which is unnecessary (e.g. what is AUC and similar)

Language and style
References are not properly formatted (ordinal number is written twice).
In some parts language is unclear (e.g. lines 61 - 62, 124-125, 225) or surplus e.g. 72-73 (road maintenance of roads)
In some parts there flow of information is mixed (e.g. current lines 69-71 should go before lines 65-68)
Table 2 - I found odd that noun "resolution" is used for the quantity which I believe is sample rate (over time or over distance).
Text formatiing errors

[1] Reference Test System for Machine Vision Used for ADAS Functions
[2] Decision Tree Method to Analyze the Performance of Lane Support Systems

Author Response

(The authors gave the same response as above.)

Round 2

Reviewer 3 Report

Now, the paper can be published.

Author Response

Thank you for reviewing our paper.

Reviewer 4 Report

N/A

Author Response

Thank you for reviewing our paper.